# The Role of Resilience, Happiness, and Social Support in the Psychological Function during the Late Stages of the Lockdown in Individuals with and without Chronic Pain

**DOI:** 10.3390/ijerph19116708

**Published:** 2022-05-31

**Authors:** Jordi Miró, Elisabet Sánchez-Rodríguez, M. Carme Nolla, Rui M. Costa, J. Pais-Ribeiro, Alexandra Ferreira-Valente

**Affiliations:** 1Universitat Rovira i Virgili, Unit for the Study and Treatment of Pain—ALGOS, Research Center for Behavior Assessment (CRAMC), Department of Psychology, 43007 Tarragona, Spain; elisabet.sanchez@urv.cat (E.S.-R.); cnolla@xarxatecla.cat (M.C.N.); 2Institut d’Investigació Sanitària Pere Virgili, Universitat Rovira i Virgili, 43007 Tarragona, Spain; 3Xarxa Social i Sanitària de Santa Tecla, 43003 Tarragona, Spain; 4William James Center for Research, Ispa—Instituto Universitário, 1149-041 Lisbon, Portugal; rcosta@ispa.pt (R.M.C.); jlpr@fpce.up.pt (J.P.-R.); mafvalente@gmail.com (A.F.-V.); 5Department of Rehabilitation Medicine, University of Washington, Seattle, WA 98109, USA

**Keywords:** COVID-19, social distancing measures, pain, happiness, social support, resilience

## Abstract

There is mounting evidence to suggest that individuals with chronic pain adjusted poorly to and were impacted negatively by social distancing measures during the lockdown. However, there is limited data on the factors that might protect against the negative effects associated with social distancing measures, as most research has been conducted in the general population and in the initial stages of the lockdown. The aim of this study was to improve the understanding of the role that resilience, happiness, and social support, all factors that are thought to have a protective role, played in the psychological function (measured as anxiety, depression, and stress) to the social distancing measures during the late stages of the lockdown in a sample of adults with and without chronic pain living in Spain. A group of 434 adults responded to an online survey and provided information on sociodemographic issues, which included measures of pain, perceived health and quality of life, depression, anxiety, stress, resilience, happiness, and social support. The data showed that individuals with chronic pain (N = 200; 46%) reported statistically significant worst psychological function, that is to say, they reported higher levels of anxiety, depression, and stress (all *p*s < 0.001). Resilience, social support, and happiness proved to be significant predictors of anxiety, depression, and stress, after controlling for the effects of age, gender, and chronic pain. Although the effect sizes were small to medium, they are consistent with the findings of other studies. The findings from this study provide important additional new information regarding the associations between resilience, happiness, and social support and the adjustment to the social distancing measures during the late stages of the lockdown. These findings can be used to develop programs to improve adjustment to and coping with the demands of social distancing measures.

## 1. Introduction

Research has shown that COVID-19 social distancing measures, despite being necessary to control pandemic impact, have been associated with an increased risk of problems for individuals with chronic pain. For example, Toorak and colleagues [1] found that patients who stayed at home during the lockdown reported higher levels of low back pain than those who carried on working outside the home. In a study with a heterogeneous sample of adults with chronic pain, Zambelli and colleagues [2] observed that during lockdown, restricted access to healthcare and increased dependence on others were associated with negative wellbeing outcomes related to sleep, anxiety, and depression. Similarly, in a study with 502 adults with chronic pain, Nieto and colleagues [3] found that half of the participants increased unsupervised medication intake, which has been found to be related to increased problems in both psychological and social function among individuals with chronic pain [4]. In a cross-sectional survey with a sample of 150 patients with fibromyalgia, chronic spine pain, and postsurgical pain, Hruschak and colleagues [5] reported that patients’ characteristics were associated with a greater impact of social distancing. Specifically, being a female and nonwhite, lower education, and higher pain catastrophizing were independently associated with greater pain severity, whereas being female and pain catastrophizing were also independently associated with greater pain interference. In the same way, Serrano-Ibáñez and colleagues [6] found a statistically significant association between changes in daily routines, pain intensity, and emotional distress. In this study, significant predictors of emotional distress were age, difficulty in accessing medical care, changes in daily routines, and diminished social support. Together these findings consistently indicate that the social distancing measures had a negative impact on the physical, psychological, and social function of individuals with chronic pain. Moreover, research has consistently identified a group of variables that are associated with the negative impact of social distancing measures on patients with chronic pain and their poor adjustment to them (e.g., catastrophic thinking, anxiety; [5,7]).

It is important to note that research has also identified factors that seem to have a protective role and help patients to adjust to the social distancing measures imposed during the lockdown. For example, Pais-Ribeiro and colleagues [8] found that satisfaction with social support from friends and close ones predicted lower levels of depression, anxiety, and stress. Similarly, Ferreira and colleagues [9] reported that positivity was a predictor of psychological well-being. In a sample of adults living in Turkey, Peker and Cengiz [10] found that resilience (i.e., the ability to thrive despite exposure to adversity [11]) was associated with fear and perceived stress, and mitigated the impact of fear on happiness. Similar studies conducted in Germany [12] and Italy [13] have shown associations between resilience, stress, and adjustment to social distancing measures. In addition, other studies have reported significant positive relations between resilience and social support [14], which in turn has been positively associated with mental health [15].

However, very little information is available about what factors might have a protective role against the negative effects associated with social distancing measures in individuals with chronic pain, as most research has focused on the general population and the initial stages of the lockdown. There is a need for research to identify those factors that help individuals with health conditions such as chronic pain, particularly those that are modifiable, to adjust to and cope with the problems associated with social distancing measures. The findings could help to develop programs that foster and reinforce protective factors to prevent or alleviate the negative consequences of social distancing measures, and improve the adjustment of individuals with and without chronic pain.

Given these considerations, the aim of this study was to improve our understanding of the role that resilience, happiness, and social support played in the psychological function (measured as depression, anxiety, and stress) to social distancing measures during the late stages of the lockdown in a sample of adults with and without chronic pain living in Spain.

## 2. Methods

### 2.1. Participants

The participants came from a sample of 544 adults who showed an interest in participating in an online survey to study the impact of COVID-19 social distancing measures during the lockdown on the life of adults living in Spain. In this study, only data from 434 potential participants that provided complete information were included. Although two articles have been published using data from this survey [16,17], neither of them addressed the question that is the focus of the current study.

The inclusion criteria for the study were that participants had to: (1) be adults living in Spain during the lockdown; (2) have access to the Internet; (3) be able to read and write Spanish; and (4) provide their informed consent.

### 2.2. Procedure

We used social media (i.e., Facebook, Instagram, Twitter) and the university community to publicize the study and share the link to the survey. When they entered the survey, the first thing participants found was a detailed explanation of the study and the informed consent page. Then, in order to participate, interested individuals had to express their consent by clicking “YES” in response to a question about consent. The online survey was available between 3 June and 30 July 2020. The study was approved by the Ethical Committee for Medical Research of the Pere Virgili Health Research Institute (ref. 117/2020).

### 2.3. Measures

Demographic and social characteristics of the sample were studied. Participants were requested to inform about their age, gender, and education.

Regarding pain, participants were asked to report whether they had experienced pain during the lockdown. Then, they were asked to report the duration of the pain problem using a 7-point verbal rating scale with alternatives ranging from “<than three months” to “≥ten years”. In this study, following the definition of the International Association for the Study of Pain [18], chronic pain has been conceptualized as pain that has been present for at least three months.

### 2.4. Outcome Variables

On the topics of depression, anxiety, and stress, we used the Spanish version of the 21-item Depression, Anxiety, and Stress Scale to measure symptoms of anxiety, depression, and stress (DASS; [19,20]). Respondents were asked to state the degree to which each symptom had been present since the start of the COVID-19 State of Emergency on 14 March. They were asked to respond on a 4-point Likert-type scale, from 0 (“Did not apply to me at all”) to 3 (“Applied to me very much or most of the time”). The total score on each scale was the sum of the responses multiplied by 2. In this study, the internal consistency was excellent for the stress scale (α = 0.90), and good for the anxiety (α = 0.83) and depression scales (α = 0.89).

### 2.5. Predictive Variables

To measure resilience, we used the 10-item Spanish version of the Connor–Davidson resilience scale (CD-RISC-10; [21,22]). With the CD-RISC-10, respondents were asked to answer, on a 5-point Likert-type scale (from 0 “Never” to 4 “Almost always”), to what extent each of the statements applies to them, in relation to what happened during the state of alarm decreed by COVID-19 on 14 March 2020. The total score is a sum of the responses (range 0–40), and higher scores indicate higher levels of resilience. In this study, the internal consistency for the scale was good (α = 0.89).

In terms of happiness, we took a question from the 2010–2012 World Values Survey [23] for participants to report on their happiness. They were given the following instructions to report on their perceived happiness: “All things considered, would you say that you are very happy, happy, a little happy or unhappy. Please indicate the answer that is best for you since the start of the COVID-19 State of Emergency on March 14”. In this study, we have reversed the order so that scores were computed in such a way that the name of the variable made sense. Thus, in this study, higher scores indicate higher levels of happiness.

For social support, we used the Intimacy scale of the Social Support Satisfaction Scale [24]. This scale has four items (e.g., “Sometimes I feel alone in the world and without support”), which correspond to a 5-point Likert scale ranging from “1 = strongly agree” to “5 = totally disagree”. Higher scores are indicative of greater intimate social support. In this study, the internal consistency for the scale was good (α = 0.75).

### 2.6. Data Analyses

We first computed the means, standard deviations, number, and percentages of the variables to describe the sample of participants. Next, we examined the data to ensure that the assumptions for the planned analysis were met. After confirming the absence of significant skewness, kurtosis, outliers, heteroscedasticity, dependence, and multicollinearity for any predictor or criterion variable (parameters: skewness < 3, kurtosis < 10, and variance inflation factors <10.0 [25], we performed *t*-tests to determine if there were statistically significant differences between participants with and without chronic pain in the study variables (i.e., resilience, happiness, social support, anxiety, depression and stress), and also computed zero-order correlations between the key predictor (resilience, happiness, and social support) and criterion (anxiety, depression, and stress) variables. Finally, we performed three hierarchical multiple regression analyses to evaluate the relative importance of resilience, happiness, and social support as predictors of anxiety, depression, and stress in the study sample. In these analyses, we first entered demographic variables (gender (coded as 0 for male and 1 for female), age and education) in step 1, and then chronic pain in step 2, to control for its potential confounding effects. In step 3, we entered the primary predictors (resilience, happiness, and social support) to determine the extent to which they contributed to the prediction of the criterion variables over and above the demographic variables and chronic pain status.

## 3. Results

### 3.1. Description of the Study Sample and the Study Variables

A total of 434 adults (average age of 38.95 years old (SD = 14.74)) responded to the online survey, provided responses to all the questionnaires, and were included in the analyses. Of these, 331 (76%) were women. Most participants had a university degree (61%). Almost half of the sample (N = 200; 46%) reported having chronic pain, defined as pain that lasts for more than 3 months.

### 3.2. Assumptions Testing

All the scores on the measures were normally distributed (skewness and kurtosis ranging from −0.58 to 1.76, and −0.47 to 3.49). Moreover, p-plots suggested the normality of the residuals and scatter plots showed a lack of heteroscedasticity. Independence was supported by Durbin–Watson statistics ranging from 1.87 to 2.02 (all of them between 1.5 and 2.5). The lack of multicollinearity was supported by VIFs lower than 10 and tolerance values close to 1 (ranging from 0.64 to 0.97). See Table 1 for additional details on the study variables.

### 3.3. Comparisons between Participants with and without Chronic Pain

Table 2 summarizes the data about the variables in the study for participants with and without chronic pain. As can be seen, participants with chronic pain reported higher levels of anxiety, depression, and stress, whereas participants without chronic pain reported higher levels of happiness, overall health, and quality of life.

### 3.4. Zero-Order Correlations between Criterion and Predictor Variables

Table 3 shows the zero-order Pearson correlation coefficients between the criterion and the predictor variables. As can be seen, all the variables were significant and at least moderately associated with each other (see Table 3).

### 3.5. Regression Analyses

#### 3.5.1. Predicting Anxiety

The regression analysis predicting anxiety, age, gender, and education explained 7% of the variance (see Table 4), largely because of the effects of age (β = −0.17, *p* < 0.001) and gender (β = 0.11, *p* < 0.01), whereas chronic pain accounted for an additional 5% of the variance in the criterion (β = 0.17, *p* < 0.001). Finally, resilience, happiness, and social support, as a block, explained an additional 15% of the variance (β = −0.14, −0.23 and −0.13, respectively, all *p*s < 0.01). In summary, younger people, females, and those who had chronic pain were more likely to report higher levels of anxiety. Furthermore, those who reported higher levels of resilience, happiness, and social support were less likely to report anxiety symptoms.

#### 3.5.2. Predicting Depression

The regression analyses predicting depression, age, gender, and education explained 8% of the variance of depression (see Table 5), largely because of the effects of age (β = −0.21, *p* < 0.001). Chronic pain accounted for an additional 3% of the variance (β = 0.11, *p* < 0.01). Finally, resilience, happiness, and social support, as a block, explained an additional 38% (β = −0.19, −0.34 and −0.26, respectively, all *p*s < 0.001). In summary, younger people and those who had chronic pain were more likely to report higher levels of depression. Furthermore, those who reported higher levels of resilience, happiness, and social support were less likely to report depressive symptoms.

#### 3.5.3. Predicting Stress

The regression analyses predicting stress, age, gender, and education explained 9% of the variance of stress (see Table 6), largely because of the effects of age and gender (β = −0.21 and 0.13, respectively, *p*s < 0.01). Chronic pain explained an additional 3% of the variance (β = 0.14, *p* < 0.001). Finally, resilience, happiness, and social support, as a block, explained an additional 17% (β = −0.13, −0.28 and −0.10, respectively, all *p*s < 0.05). In summary, younger people, females, and those who had chronic pain were more likely to report higher levels of stress. Furthermore, those who reported higher levels of resilience, happiness, and social support were less likely to report stress.

## 4. Discussion

The aim of this study was to improve our understanding of the role that resilience, happiness, and social support play in the psychological function (measured as anxiety, depression, stress, perceived overall health, and quality of life) during the late stages of the COVID-related lockdown among adults with and without chronic pain.

Two key findings emerged. First, the data showed that individuals with and without chronic pain differed in important ways. That is, participants with chronic pain reported statistically significant higher levels of anxiety, depression, and stress. These findings are consistent with studies showing that individuals with chronic pain experience statistically and significantly worse psychological function [26,27]. In fact, similar to previous studies, including those conducted during the first-wave lockdown (e.g., [5,28,29,30]), the data also showed that chronic pain is a significant predictor of worse psychological function.

Second, even though the data showed no differences in terms of resilience and social support between the two groups, these two variables and happiness were found to be significant predictors of anxiety, depression, and stress, after controlling for the effects of age, gender, and chronic pain. Although the effect sizes were small to medium, they are consistent with the findings reported in studies examining the associations between resilience, happiness, and social support and psychological function in individuals with chronic pain [31]. For example, research has shown that resilience, measured as psychological flexibility [32], significantly predicted pain interference and depression after adjusting for age, pain, and anxiety, in a sample of adults with chronic pain. Similar results have been reported in studies with elderly patients [33] and adolescents [34], and with samples of individuals without chronic pain (e.g., [35]). Future research should study the factors that are related to resilience, in particular those that are modifiable and could be used to increase resilience (e.g., emotional support systems [36]), help to reduce disability, and improve function.

There were no differences in terms of social support reported by participants with and without chronic pain in this study. However, social support proved to be a significant predictor of psychological function (i.e., anxiety, depression, and stress). These findings are in line with research showing that social support predicts mood and function in studies conducted with samples of adults with chronic pain (e.g., [37,38,39]) and without (e.g., [40,41,42]). Research on social support in individuals with chronic pain has shown that social distancing measures are associated with increased loneliness [43], reduced access to high-quality pain management [2], and inequalities of different types (e.g., reduced economic resources [29]).

In this study, the data also showed that happiness proved to be a predictive factor of psychological function (i.e., anxiety, depression, and stress). These findings are also consistent with previous research that showed happiness to be related to lower pain interference and distress [44]. Moreover, in this sample, happiness was found to be a significant predictor of psychological function regardless of the chronic pain status of the participants. If replicated, this finding would suggest that happiness might act as protective factor in individuals who are subjected to social distancing measures over long periods. To the extent that the association was found to be causal—something that would have to be specifically examined in future studies—the finding would suggest that programs targeting happiness could also expect to improve psychological, physical, and social function. Although happiness might be a difficult variable to change or improve, it would be important to identify the factors that contribute to happiness in these circumstances and study their potential value as resources to improve function. In this sample, happiness was associated with social support, and this has been found as a protective factor of psychological, physical, and social function [39]. Identifying specific targets for intervention is a valuable step in the development and implementation of effective strength- and resource-based interventions for future similar situations in which social distancing measures must be implemented.

Although this study offers valuable information about the role of some psychosocial variables (i.e., resilience, happiness, and social support) in the psychological function of individuals with and without chronic pain, to the social distancing measures during the lockdown, it has limitations that should be considered when interpreting the results. First, although the number of participants was appropriate for the planned analysis, the sample size was relatively small. Therefore, new studies with larger sample sizes, including other variables (e.g., income measurements, computer skills), would be necessary to determine the reliability and validity of the findings. Second, the cross-sectional design of the study did not enable causal associations between the variables to be evaluated, and only concurrent associations were studied. Future studies with longitudinal designs are warranted. Third, although the procedure is widely used and found reliable for the most part, we cannot overrule issues in relation to inclusion criteria. For example, it is difficult to verify that all participants complied with the inclusion criteria. Nevertheless, if there was any failure to comply with the study conditions, it probably was to a very little extent. Thus, additional studies would be needed to develop data-based predictions, including studies with samples with different chronic conditions.

## 5. Conclusions

Despite the study’s limitations, the findings provide important additional new information regarding the associations between resilience, happiness, and social support and psychological function (i.e., anxiety, depression, and stress) in adults with and without chronic pain during the late stages of the lockdown. The two main findings were that: (1) individuals with and without chronic pain differed in important ways. That is, participants with chronic pain showed a worse psychological function (i.e., they reported statistically significant higher levels of anxiety, depression, and stress); and (2) happiness, resilience, and social support were found to be significant predictors of psychological function (i.e., anxiety, depression, stress) after controlling for the effects of age, gender, and chronic pain. The findings are consistent with the idea that although social distancing measures are needed to control the pandemic, they also have negative outcomes. Moreover, and more importantly, some protective factors can be used to improve psychological function. The findings from this study can be used to inform policy and specific responses to future COVID-19 waves and pandemics. This is key as new waves are already swamping some world regions, and an increase in chronic pain and related disabilities following the COVID-19 pandemic is expected.

## Figures and Tables

**Table 1 ijerph-19-06708-t001:** Description of the study variables.

	Mean	SD	Range	Skewness	Kurtosis
Resilience	26.92	7.43	2–40	−0.37	−0.13
Happiness	2.85	0.59	1–4	−0.49	1.02
Social Support	14.98	4.10	4–20	−0.58	−0.47
Anxiety	5.94	7.10	0–42	1.76	3.49
Depression	7.85	8.61	0–42	1.56	2.27
Stress	12.39	9.64	0–42	0.87	0.18

**Table 2 ijerph-19-06708-t002:** Mean comparisons between participants with and without chronic pain.

	NCP ($\bar{x}$) N = 234	CP ($\bar{x}$) N = 200	*t*-Test	d Cohen
Resilience	27.23	26.56	0.93	0.09
Happiness	2.93	2.76	3.09 *	0.30
Social Support	15.23	14.70	1.35	0.13
Anxiety	4.54	7.57	−4.43 **	0.43
Depression	6.55	7.21	−3.37 **	0.33
Stress	10.73	14.34	−3.89 **	0.37

Note: NCP: Participants without chronic pain; CP: Participants with chronic pain. * *p* < 0.01, ** *p* < 0.001.

**Table 3 ijerph-19-06708-t003:** Correlations between criterion and predictor variables.

	Anxiety	Depression	Stress	Resilience	Happiness
Resilience	−0.35 *	−0.50 *	−0.36 *		
Happiness	−0.40 *	−0.58 *	−0.43 *	0.50 *	
Social Support	−0.32 *	−0.50 *	−0.30 *	0.33 *	0.46 *

Note: * *p* < 0.001.

**Table 4 ijerph-19-06708-t004:** Regression analyses predicting anxiety.

	Model 1			Model 2			Model 3		
Predictor	β	*t*	*p*	β	*t*	*p*	β	*t*	*p*
Demographic variables									
Age	−0.17	30.66	<0.001	−0.21	4.46	<0.001	−0.17	3.90	<0.001
Gender ^+^	0.18	30.80	<0.001	0.13	2.84	0.005	0.11	2.63	0.009
Education	−0.09	10.82	0.070	−0.07	1.59	0.112	−0.03	−0.65	0.514
Chronic pain				0.22	4.72	<0.001	0.17	3.94	<0.001
Protective variables									
Resilience							−0.14	2.75	0.006
Happiness							−0.23	4.52	<0.001
Social Support							−0.13	2.81	0.005
	R^2^ = 0.07			R^2^ = 0.12			R^2^ = 0.27		
	R^2^ change = 0.07			R^2^ change = 0.05			R^2^ change = 0.15		
	F = 110.45			F = 220.23			F = 300.07		
	*p* < 0.001			*p* < 0.001			*p* < 0.001		

Note: ^+^ Dummy coded as 0 male and 1 female. Highlighted *p*s were significant at 0.01, or 0.001.

**Table 5 ijerph-19-06708-t005:** Regression analyses predicting depression.

	Model 1			Model 2			Model 3		
Predictor	β	*t*	*p*	β	*t*	*p*	β	*t*	*p*
Demographic variables									
Age	−0.24	50.08	<0.001	−0.27	5.75	<0.001	−0.21	5.78	<0.001
Gender ^+^	0.11	20.34	0.020	0.07	1.50	<0.135	0.04	1.13	0.258
Education	−0.10	20.14	0.033	−0.09	1.95	0.052	−0.02	0.46	0.647
Chronic pain				0.19	4.04	<0.001	0.11	3.08	0.002
Protective variables									
Resilience							−0.19	4.74	<0.001
Happiness							−0.34	7.95	<0.001
Social Support							−0.26	6.60	<0.001
	R^2^ = 0.08			R^2^ = 0.11			R^2^ = 0.49		
	R^2^ change = 0.08			R^2^ change = 0.03			R^2^ change = 0.38		
	F = 130.25			F = 160.28			F = 1070.7		
	*p* < 0.001			*p* = 0.002			*p* < 0.001		

Note: ^+^ Dummy coded as 0 male and 1 female. Highlighted *p*s were significant at 0.01, or 0.001.

**Table 6 ijerph-19-06708-t006:** Regression analyses predicting stress.

	Model 1			Model 2			Model 3		
Predictor	β	*t*	*p*	β	*t*	*p*	β	*t*	*p*
Demographic variables									
Age	−0.22	40.71	<0.001	−0.25	5.42	<0.001	−0.21	5.04	<0.001
Gender^+^	0.19	0.19	<0.001	0.15	3.26	0.001	0.13	3.09	0.002
Education	−0.02	0.44	0.662	−0.01	0.21	0.835	0.03	0.83	0.410
Chronic pain				0.20	4.19	<0.001	0.14	3.30	<0.001
Protective variables									
Resilience							−0.13	2.62	0.009
Happiness							−0.28	5.55	<0.001
Social Support							−0.10	2.19	0.029
	R^2^ = 0.09			R^2^ = 0.12			R^2^ = 0.29		
	R^2^ change = 0.09			R^2^ change = 0.03			R^2^ change = 0.17		
	F = 140.15			F = 170.52			F = 330.92		
	*p* < 0.001			*p* =< 0.001			*p* < 0.001		

Note: ^+^ Dummy coded as 0 male and 1 female. Highlighted *p*s were significant at 0.01, or 0.001.

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
