# Peer review of "The Role of Resilience, Happiness, and Social Support in the Psychological Function during the Late Stages of the Lockdown in Individuals with and without Chronic Pain"

_ijerph, 2022, doi:10.3390/ijerph19116708_

Round 1

Reviewer 1 Report

The authors sufficiently addressed my previous concerns, and I have nothing more to add. The paper will make a nice contribution toward understanding pandemic experiences of people managing chronic pain.

Author Response

Authors’ response: We thank the reviewer for the kind comments.

Reviewer 2 Report

the version offered for review was the one full of corrections: this makes it difficult for me to concentrate on evaluating. The item is interesting and the sample size sufficient for a pilot study. I suggest the following review points which would be enough to move on to publication:

1.       I think it is necessary to better define the concept of resilience since it is an important reference for all the work. you can see how much in https://pubmed.ncbi.nlm.nih.gov/31744109/ (but it is not necessary to mention it).

2.       the procedure (see point 2.2.) seems unreliable to me because it makes it difficult to verify the inclusion criteria. I think it may be a weakness that the authors should declare.

3.       It does not seem possible to assume that the results are the result of the lockdown. to reach this conclusion it would be useful (at least in the final intentions) to envisage a study on a similar population without the lockdown regime and thus to be able to compare the results

4.       It does not seem to me that the authors take into consideration the degree of culture and computer skills of the people involved. I think this is an important fact to evaluate.

5.       in the same way, I see no reference to the state of nutrition or eating habits that can be relevant elements both for the PMI and for the production of endorphins or for nutritional principles such as Vitamin B

Author Response

the version offered for review was the one full of corrections: this makes it difficult for me to concentrate on evaluating.

Authors’ response: We apologize for this. It seems that we mistakenly uploaded a wrong file. We are very sorry for the additional time and effort that this has caused you in reviewing our paper.

 The item is interesting and the sample size sufficient for a pilot study.

Authors’ response: We thank the reviewer for the kind comments.

I suggest the following review points which would be enough to move on to publication:

  1. I think it is necessary to better define the concept of resilience since it is an important reference for all the work. you can see how much in https://pubmed.ncbi.nlm.nih.gov/31744109/(but it is not necessary to mention it).

Authors’ response: A definition of what resilience is and a citation to support this is provided in page 1 [Cusinato, M.; Iannattone, S.; Spoto, A.; Poli, M.; Moretti, C.; Gatta, M.; Miscioscia, M. Stress, Resilience, and Well-Being in Italian Children and Their Parents during the  COVID-19 Pandemic. Int. J. Environ. Res. Public Health 2020, 17.].

  1. the procedure (see point 2.2.) seems unreliable to me because it makes it difficult to verify the inclusion criteria. I think it may be a weakness that the authors should declare.

Authors’ response: Done as requested (see page 7).

  1. It does not seem possible to assume that the results are the result of the lockdown. to reach this conclusion it would be useful (at least in the final intentions) to envisage a study on a similar population without the lockdown regime and thus to be able to compare the results

Authors’ response: The reviewer has a good point here. This was a cross-sectional study. Therefore, it precludes any conclusions regarding causative relationships. In this study, we report on the association among the variables during the lockdown. We have revised the limitations section and added the importance of longitudinal studies in future research (see page 7).

  1. It does not seem to me that the authors take into consideration the degree of culture and computer skills of the people involved. I think this is an important fact to evaluate.

Authors’ response: The reviewer is correct in this. We did not control for the influence of these variables. It is unclear whether they could have any influence on the results, and then, if any, in what direction. We address this as a limitation to our study. We have revised this section to add the variable “computer skills” as a focus of interest in future studies (see page 7).

  1. in the same way, I see no reference to the state of nutrition or eating habits that can be relevant elements both for the PMI and for the production of endorphins or for nutritional principles such as Vitamin B

Authors’ response: It is unclear what is meant here. We do not know what “PMI” stands for, or what interest may have “nutrition” in this particular study. We would be happy to address this if provided additional guidance.

This manuscript is a resubmission of an earlier submission. The following is a list of the peer review reports and author responses from that submission.

Round 1

Reviewer 1 Report

The article takes up an important and current topic. However, both in theoretical and practical terms, it is not good enough and in current form is not suitable for publication.

The theoretical part is very week. There is actually no literature review section. The selection of variables in the study was not justified any way (is happiniess really predictor of depression or maybe lack of happiness is an effect of depression?).

It has not been explained what is the added value of the article.

The sample is unbalanced with the overrepresentation of women (76% of women) and, as the description of the sample selection shows, it is unrepresentative, so it cannot be used to draw general conclusions.

The statistical methods used are also inadequate. It cannot be said that variables measured on the Likert scale are normally distributed, because the Likert scale is a weak (Ordinal) scale and the normal distribution applies to variables measured on the ratio scale. What is more t-test cannot be applied, because it is dedicated for variables measured on ratio scale. The conclusions drawn from the t-tests are trivial (“participants without chronic pain reported higher levels of happiness, overall health, and quality of life”).

Regression function is not clearly described (what kind of regression was used, what is reference value of gender). Obtained parameters were not interpreted.

Summing up, I regret to say that the presented work does not meet the criteria of a scientific article.

Author Response

The article takes up an important and current topic. However, both in theoretical and practical terms, it is not good enough and in current form is not suitable for publication.

The theoretical part is very week. There is actually no literature review section.

Authors’ response:  We would be happy to edit the contents of the Introduction to improve it, but it is unclear what the specific issues are, based on this comment. In the Introduction, we provided a review of the findings on the effects of social distancing measures in individuals with and without chronic pain. In doing so, we reported on the factors that have been found to be significantly associated with increased problems (i.e., risk factors). We also reported on the very few studies on potential protective factors. The findings from these studies are the basis of our study. We have edited the last paragraph of the Introduction to make this clearer to readers (see page 5).  

The selection of variables in the study was not justified any way (is happiniess really predictor of depression or maybe lack of happiness is an effect of depression?).

Authors’ response:  The selection of variables was justified on the data available that had been briefly summarized in the Introduction section. This was a cross-sectional study, therefore we cannot draw any conclusions from our data about causality. This is a limitation to the study (as it is the case of all cross-sectional studies), and we describe it in the limitations section of our manuscript (see page 20).

It has not been explained what is the added value of the article.

Authors’ response:  The need and interest of this study was described at the end of page 4 and beginning of page 5, right above the last paragraph of the Introduction, before the description of the objectives. The value of the findings of this study and implications are also described in the Discussion section (see pages 17-20). 

The sample is unbalanced with the overrepresentation of women (76% of women) and, as the description of the sample selection shows, it is unrepresentative, so it cannot be used to draw general conclusions.

Authors’ response:  There were more women participating in this study. Generally speaking, this might be a problem to draw general conclusions. However, it is unclear if this was really the case in our study. The fact of the matter is that the data and findings in our study were very similar to published studies. Therefore, this is somewhat of a validity check to our study and findings. Nevertheless, in the Discussion, when we allude to the study limitations, we describe the need for additional studies to determine the reliability and validity of the findings.

In addition, it is important to note that we did not select the participants in this study, as they were volunteer individuals participating in a study about the impact of lock down measures. From the sample of volunteers we “selected” those that complied with the inclusion criteria for this study, as described in page 5. We realize, however, that the way in which we described the procedure might inadvertently contribute to this view. Therefore we changed the words used and replaced “were selected” by “came” (see page 5).  

The statistical methods used are also inadequate. It cannot be said that variables measured on the Likert scale are normally distributed, because the Likert scale is a weak (Ordinal) scale and the normal distribution applies to variables measured on the ratio scale. What is more t-test cannot be applied, because it is dedicated for variables measured on ratio scale.

Authors’ response:  The Reviewer is correct in calling our attention to the need of using non-parametric tests with ordinal variables. However, simulation research has shown that results of parametric and non-parametric statistics are similar when using ordinal variables with at least 5 levels (at least 5-point likert scales), and given that parametric statistics have a higher degree of statistical power, it is a common practice to use parametric statistics with this kind of variables. Therefore, in using t-tests, we followed this common practice [see Michell, J. (1990). An Introduction to the Logic of Psychological Measurement. Hillsdale, NJ: Erlbaum; Michell, J. (1997). Quantitative science and the definition of measurement in psychology. British Journal of Psychology, 88, 355- 383. doi:org/10.1111/j.2044-8295.1997.tb02641.x; Michell, J. (1999). Measurement in Psychology: A Critical History of a Methodological Concept. New York, NY: Cambridge University Press.: Michell, J. (2000). Normal science, pathological science and psychometrics. Theory & Psychology, 10, 639–667. doi: 10.1177/0959354300105004; Michell, J. (2002). Stevens's theory of scales of measurement and its place in modern psychology. Australian Journal of Psychology, 54, 99 – 104. doi:10.1080/00049530210001706563; Michell, J. (2005). The logic of measurement: A realist overview. Measurement, 38, 285–294. doi:10.1016/j.measurement.2005.09.004; Michell, J. (2008). Is psychometrics pathological science? Measurement, 6, 7–24. doi:10.1080/15366360802035489; Michell, J. (2009). The psychometricians’ fallacy: too clever by half? British Journal of Mathematical and Statistical Psychology, 62, 41–55. doi:org/10.1348/000711007X243582; Michell, J. (2013). Constructs, inferences, and mental measurement. New Ideas in Psychology, 31, 13–21. doi:org/10.1016/j.newideapsych.2011.02.004.]  Moreover, we would like to highlight that we have now conducted a new set of statistical analysis using Mann-Whitney U test and the results have been essentially the same.

The conclusions drawn from the t-tests are trivial (“participants without chronic pain reported higher levels of happiness, overall health, and quality of life”).

Authors’ response:  We described the result of the analysis, we did not conclude anything in the Results section. It is unclear what the Reviewer is suggesting us to do here, if anything. The conclusions of our study are described in pages 17-20.

Regression function is not clearly described (what kind of regression was used, what is reference value of gender).

Authors’ response:  We have clarified this issue in the manuscript (see page 8). We conducted hierarchical regression analysis. The variable gender was dummy coded as 0 for males and 1 for females

Obtained parameters were not interpreted.

Authors’ response:  In the Results section, we described the result of the analysis; we did not elaborate on the conclusions or interpret the findings. It is unclear what the Reviewer is suggesting us to do here, if anything. The conclusions of our study are described in pages 17-20.

Summing up, I regret to say that the presented work does not meet the criteria of a scientific article.

Authors’ response:  We hope that our explanations and revisions to the original manuscript adequately address the concerns raised by the Reviewer. 

Reviewer 2 Report

Dear authors,

the resilience is a great theme nowadays because it is something that we should have. 

I think that the idea that you give is quite interesting for having new programs which should be attended this. Although, I have to admit that there has been some programs which are currently interested in that topic. 

Thanks very much for giving the opportunity for reading this paper. 

It was a great opportunity to read your article and I will suggest:

  • If you could do a shorter title of the article
  • Define the concept of resilience of education to facilitate new programs about it
  • Try to incorporate some of these articles in your paper
  • Sampogna, G. et al. What is the role of resilience and coping strategies on the mental health of the general population during the COVID-19 pandemic? Results from the Italian Multicentric COMET Study. Brain Sci. 11, 1231 (2021). 21.
  • Rice, K., Rock, A. J., Murrell, E. & Tyson, G. A. The prevalence of psychological distress in an Australian TAFE sample and the relationships between psychological distress, emotion-focused coping and academic success. Aust. J. Psychol. 73, 231–242 (2021). 22.
  • StanisÅ‚awski, K. The coping circumplex model: An integrative model of the structure of coping with stress. Front. Psychol. 10, 694 (2019). 23.
  • Rodríguez-Rey, R., Garrido-Hernansaiz H & Collado, S. (2020). Psychological Impact and Associated Factors During the Initial Stage of the Coronavirus (COVID-19) Pandemic Among the General Population in Spain. Front. Psychol. 11:1540. doi: 10.3389/fpsyg.2020.01540
  • Violant-Holz, V., Gallego-Jiménez, M. G., González-González, C. S., Muñoz-Violant, S., Rodríguez, M. J., Sansano-Nadal, O., & Guerra-Balic, M. (2020). Psychological Health and Physical Activity Levels during the COVID-19 Pandemic: A Systematic Review. International journal of environmental research and public health, 17(24), 9419.
  • Muñoz-Violant, S., Violant-Holz, V., Gallego-Jiménez, M. G., Anguera, M. T., & Rodríguez, M. J. (2021). Coping strategies patterns to buffer the psychological impact of the State of Emergency in Spain during the COVID-19 pandemic’s early months. Scientific reports, 11(1), 1-16.

Author Response

Dear authors,

the resilience is a great theme nowadays because it is something that we should have. 

I think that the idea that you give is quite interesting for having new programs which should be attended this.

Authors’ response:  We thank the reviewer for the kind comment.

Although, I have to admit that there has been some programs which are currently interested in that topic. 

Thanks very much for giving the opportunity for reading this paper. 

It was a great opportunity to read your article and I will suggest:

  • If you could do a shorter title of the article

Authors’ response:  We concur that, in general, short titles are better than long ones. Nevertheless, this time, we thought it would be better to compromise this general rule to be as informative as possible about the objectives and findings of our study, even if this meant to have a long(er) title.

  • Define the concept of resilience of education to facilitate new programs about it

Authors’ response:  We provided a definition of resilience in the manuscript (please see page 4). It is unclear what it is meant here. We do not use the concept “resilience of education”. We would be happy to address this suggestion, but would need additional details to identify what the issue is.

  • Try to incorporate some of these articles in your paper
  • Sampogna, G. et al. What is the role of resilience and coping strategies on the mental health of the general population during the COVID-19 pandemic? Results from the Italian Multicentric COMET Study. Brain Sci. 11, 1231 (2021). 21.
  • Rice, K., Rock, A. J., Murrell, E. & Tyson, G. A. The prevalence of psychological distress in an Australian TAFE sample and the relationships between psychological distress, emotion-focused coping and academic success. Aust. J. Psychol. 73, 231–242 (2021). 22.
  • StanisÅ‚awski, K. The coping circumplex model: An integrative model of the structure of coping with stress. Front. Psychol. 10, 694 (2019). 23.
  • Rodríguez-Rey, R., Garrido-Hernansaiz H & Collado, S. (2020). Psychological Impact and Associated Factors During the Initial Stage of the Coronavirus (COVID-19) Pandemic Among the General Population in Spain. Front. Psychol. 11:1540. doi: 10.3389/fpsyg.2020.01540
  • Violant-Holz, V., Gallego-Jiménez, M. G., González-González, C. S., Muñoz-Violant, S., Rodríguez, M. J., Sansano-Nadal, O., & Guerra-Balic, M. (2020). Psychological Health and Physical Activity Levels during the COVID-19 Pandemic: A Systematic Review. International journal of environmental research and public health, 17(24), 9419.
  • Muñoz-Violant, S., Violant-Holz, V., Gallego-Jiménez, M. G., Anguera, M. T., & Rodríguez, M. J. (2021). Coping strategies patterns to buffer the psychological impact of the State of Emergency in Spain during the COVID-19 pandemic’s early months. Scientific reports, 11(1), 1-16.

 Authors’ response:  We thank for this suggestion. All of the suggested studies are related to the early stages of the lockdown and none in relation to chronic pain samples or individuals suffering from chronic health conditions. One of them, the one that deals with resilience, a key variable of our study, was already cited and referenced in the original version of the manuscript (see reference number 13, pages 4 and 20).

Reviewer 3 Report

The authors examine relationships between resilience, happiness and social support and mental health outcomes under covid-responsive, social-distancing measures – a period of acute stress for many people. The paper’s unique contribution to previous pandemic-related research is its examination of the potential protectors against negative health outcomes for people experiencing chronic pain. Below I offer a few suggestions aimed at improving the paper’s impact. Following these, I recommend some additional minor edits.

  1. In the introduction and/or conclusion, a justification for why this research is needed in terms of future practice would emphasize the paper’s relevance. While I understand there is a gap in the previous research, if social distancing measures are discontinued, why should we be concerned about adjustments to them? Arguably, social distancing measures were time-limited responses to a crisis. Therefore, is adjustment to the pandemic only a temporary problem?
  1. The findings could be more clearly stated. Part of the problem may be the inclusion of so many outcome variables that are not clearly identified as such. (a) Therefore, the paper should make great effort to clearly present the outcomes under investigation, clearly stating that there are three: depression, stress and anxiety. Right now, the methods section does not distinguish between explanatory and outcomes variables. (b) It is confusing that happiness is presented as both a predictor and outcome variable. For example, the abstract states that people without chronic pain “report significantly higher levels of happiness,” but in the discussion, happiness is presented as an intervention in negative health outcomes during the lockdown. (c) In the abstract, it is implied that the dependent variable is “the adjustment to the social distancing measures during the late stages of the lockdown.” I don’t think this is the case. Rather, the outcomes are mental health indicators (not adjustments). The reference to (vague) adjustments is made again in the introduction. (d) The conclusion needs more clarity. What are the one or two takeaway points the authors want to leave the reader?
  1. Is it possible for the authors to include an income measure for its demographic controls? I would imagine that economic hardship during the pandemic may contribute to depression, stress and anxiety.

Minor suggestions:

  1. The introductory paragraph could be reorganized to emphasize that the pandemic and associated disease containment measures have been particularly hard on people experiencing chronic pain.
  1. In the Methods section, please repeat the number of participants in the online survey. Were participants thrown-out of analysis? If so, why? And how many?
  1. In Table 1, it would be helpful to have a column describing the variable, so that the reader doesn’t need to refer back to the text to understand what the range of values are and what they represent.
  1. In general, the tables are very difficult to read (it could be the double-spacing or different disciplinary practices, but maybe not). One of the more notable examples: the table’s note in Table 7 appears in a final column rather than below the table. Referring to previous publications in this journal would provide helpful examples of how to present results tables.

Author Response

The authors examine relationships between resilience, happiness and social support and mental health outcomes under covid-responsive, social-distancing measures – a period of acute stress for many people. The paper’s unique contribution to previous pandemic-related research is its examination of the potential protectors against negative health outcomes for people experiencing chronic pain. Below I offer a few suggestions aimed at improving the paper’s impact. Following these, I recommend some additional minor edits.

  1. In the introduction and/or conclusion, a justification for why this research is needed in terms of future practice would emphasize the paper’s relevance. While I understand there is a gap in the previous research, if social distancing measures are discontinued, why should we be concerned about adjustments to them? Arguably, social distancing measures were time-limited responses to a crisis. Therefore, is adjustment to the pandemic only a temporary problem?

Authors’ response:  In the last paragraph in page 4 (which continues onto page 5), we described the gap in the literature, and how filling in this gap would help to advance our knowledge to be used to deal with future lock down measures (in future crises). The findings, as described in the manuscript, can be used to inform policy and specific responses for future COVID-19 waves, and new future pandemics. This is particularly important as new waves are already swamping some world regions, and an increase in chronic pain after the COVID-19 pandemic is expected (this is key and has also been added to the manuscript (see page 18).

  1. The findings could be more clearly stated. Part of the problem may be the inclusion of so many outcome variables that are not clearly identified as such.

(a) Therefore, the paper should make great effort to clearly present the outcomes under investigation, clearly stating that there are three: depression, stress and anxiety. Right now, the methods section does not distinguish between explanatory and outcomes variables.

Authors’ response:  On page 5 we described the objectives of the study: “to improve our understanding of the role that resilience, happiness and social support played in the adjustment to social distancing measures during the late stages of the lockdown in a sample of adults with and without chronic pain living in Spain”. However, we realize that it would help to improve understand our study stating how adjustment was measured. Therefore, following this suggestion, we have specifically stated the name of the outcome variables in the description of the objectives (see page 5). In addition we have clearly identified the outcome variables separated from the predictive variables (see pages 6 and 7).  

(b) It is confusing that happiness is presented as both a predictor and outcome variable. For example, the abstract states that people without chronic pain “report significantly higher levels of happiness,” but in the discussion, happiness is presented as an intervention in negative health outcomes during the lockdown.

Authors’ response:  Following this suggestion, in order to avoid potential misunderstandings, we have deleted this information from the Abstract.

(c) In the abstract, it is implied that the dependent variable is “the adjustment to the social distancing measures during the late stages of the lockdown.” I don’t think this is the case. Rather, the outcomes are mental health indicators (not adjustments). The reference to (vague) adjustments is made again in the introduction.

Authors’ response:  Again, the Reviewer has a good point here. We assessed anxiety, depression and stress, which are “mental health indicators” or measures of “psychological function”. We also assessed perceived overall health and quality of life, which are also variables related to function and adjustment. It is to be expected that individuals with lower anxiety, depression and stress, and higher scores in overall health and quality of life are those that are adjusting better to the social distancing measures. We decided to use the expression “adjustment to COVID-19 social distancing measures”, both in the title and in the text of the manuscript, as a way to summarize what we were assessing and reporting. However, in hindsight, we realize that this might be a problem. Therefore, we have thoroughly revised the manuscript and edited the content for clarity.

(d) The conclusion needs more clarity. What are the one or two takeaway points the authors want to leave the reader?

Authors’ response:  The main information (the two takeaway points) was described in the Discussion section (see pages 17-20). There, future readers will find a description of the main findings and their implications. Nevertheless, in this revision, following the suggestion of the Reviewer we have edited the Conclusions to clarify the main takeaway points of the study (see page 20).

  1. Is it possible for the authors to include an income measure for its demographic controls? I would imagine that economic hardship during the pandemic may contribute to depression, stress and anxiety.

Authors’ response: Unfortunately, it is not possible to include this information for this group of participants. But this is a limitation, and has been added to the Discussion, with a suggestion to add this variable in future research (see page 18).

Minor suggestions:

  1. The introductory paragraph could be reorganized to emphasize that the pandemic and associated disease containment measures have been particularly hard on people experiencing chronic pain.

Authors’ response:  Done as suggested (see page 3).

  1. In the Methods section, please repeat the number of participants in the online survey. Were participants thrown-out of analysis? If so, why? And how many?

Authors’ response:  Done as suggested (see page 5).

  1. In Table 1, it would be helpful to have a column describing the variable, so that the reader doesn’t need to refer back to the text to understand what the range of values are and what they represent.

Authors’ response:  Table 1 already had a column giving the information about the range of values for each variable.

  1. In general, the tables are very difficult to read (it could be the double-spacing or different disciplinary practices, but maybe not). One of the more notable examples: the table’s note in Table 7 appears in a final column rather than below the table. Referring to previous publications in this journal would provide helpful examples of how to present results tables.

Authors’ response:  The tables have been edited to improve readability.

Round 2

Reviewer 1 Report

There were no major changes made after first review so I maintain my decision to reject the manuscript.